# Time-Course and Tissue-Specific Molecular Responses to Acute Thermal Stress in Japanese Mantis Shrimp *Oratosquilla oratoria*

**DOI:** 10.3390/ijms241511936

**Published:** 2023-07-26

**Authors:** Liwen Zhang, Zhongli Sha, Jiao Cheng

**Affiliations:** 1Laboratory of Marine Organism Taxonomy and Phylogeny, Qingdao Key Laboratory of Marine Biodiversity and Conservation, Institute of Oceanology, Chinese Academy of Sciences, Qingdao 266071, China; zhangliwen@qdio.ac.cn; 2Laboratory for Marine Biology and Biotechnology, Qingdao National Laboratory for Marine Science and Technology, Qingdao 266237, China; 3Shandong Province Key Laboratory of Experimental Marine Biology, Institute of Oceanology, Chinese Academy of Sciences, Qingdao 266071, China; 4University of Chinese Academy of Sciences, Beijing 100049, China

**Keywords:** climate change, acute heat stress, crustacean, thermal sensitivity, transcriptomics

## Abstract

Current understanding of adaptability to high temperatures is increasingly important as extreme weather events that can trigger immediate physiological stress in organisms have occurred more frequently. Here, we tracked transcriptomic responses of gills, hepatopancreas, and muscle to acute thermal exposure at 30 °C for 0.5, 6, and 12 h in an economically important crustacean, *Oratosquilla oratoria*, to gain a preliminary understanding of the tissue-specific and dynamic physiological regulation process under acute heat stress. The unique physiological responses of muscle, hepatopancreas, and gills to acute thermal stress were associated with protein degradation, lipid transport, and energy metabolism in *O. oratoria*, respectively. Functional enrichment analysis of differentially expressed transcripts and heat-responsive gene clusters revealed a biphasic protective responsiveness of *O. oratoria* developed from the early responses of signal transduction, immunity, and cytoskeleton reorganization to the responses dominated by protein turnover and energy metabolism at the mid-late stages under acute heat stress. Noteworthy, trend analysis revealed a consistently upregulated expression pattern of high molecular weight heat shock protein (HSP) family members (HSP60, HSP70, and HSP90) during the entire thermal exposure process, highlighting their importance for maintaining heat resistance in *O. oratoria*. Documenting the whole process of transcriptional responses at fine temporal resolution will contribute to a far-reaching comprehension of plastic responses to acute heat stress in crustaceans, which is critical in the context of a changing climate.

## 1. Introduction

Nowadays, rapid changes in environmental conditions due to global climate change are having profound effects on ecosystems and associated species [1,2]. Thus, one of the major emerging questions in ecology and evolution is whether and how species can cope with accelerated environmental challenges [3,4,5]. In addition to the increases in average sea temperature due to climate change over longer timescales, some extreme thermal events (e.g., marine heatwaves) that generate acute heat stress can have significant impacts on many habitats [6,7,8]. Therefore, the capability of aquatic ectotherms to cope with complex thermal challenges depends not only on their thermal acclimation capacity but also on their ability to respond rapidly to acute heat stress, which is driven in part by underlying physiological mechanisms [9,10].

In addition to species’ range and life history shifts, molecular responses (plastic and genetic variation) are both important for organisms to respond to and survive in changing environments [11,12]. Considering rapid changes in environmental conditions, plastic variation in genes could increase the chance of species’ survival through underlying phenotypic changes, such as shifts in gene regulation, and allow more time for selection to act on genetic variation [13,14]. Therefore, unraveling the underpinnings of phenotypic plasticity and environmental adaptation is critical to understanding species’ persistence under global climate change both now and into the future, especially for aquatic ectothermic animals. 

The transcriptional plasticity that controls gene expression in organisms is remarkably flexible, constantly reconfiguring itself to respond and adapt to perturbations [13]. Increasing investigations of phenotypic plasticity have revealed that numerous aquatic ectotherms can minimize acute thermal damage through multiple cellular processes, including the involvement of molecular chaperones, cell signaling, immune responses, cytoskeletal organization, energy metabolism, and protein turnover [15,16,17,18]. However, most studies only provided a generalized snapshot of molecular responses by comparing samples between two statuses (normal temperature vs. high temperature) at a single time point. In addition, landscape transcriptomic studies have suggested that tissue-specific differences in gene expression can be substantial, so it is also necessary to have a preliminary understanding of differential responses to stress in various tissues [19]. Conclusively, dissecting the dynamic and tissue-specific gene expression pattern is a fundamental challenge to deciphering the underlying adaptive mechanisms to acute environmental stress in aquatic ectotherms.

The Japanese mantis shrimp, *Oratosquilla oratoria* De Haan, 1844, is an important crustacean species widely distributed in the Northwestern Pacific Ocean and has high practical value [20]. On the one hand, mantis shrimp occupies the middle tropic level in the food chain and is an essential component of marine ecosystems [21]. They are benthic animals that help improve water quality by consuming organic matter in the sediment and removing pollutants, thus contributing to the maintenance of marine ecological environments [22]. Moreover, due to its wide latitudinal range from 15 to 45° N and biogeographical line along the Yangtze River in China, *O. oratoria*’s populations spread along a broad gradient of mean annual sea temperature [23,24]. On long-term scales, this temperature gradient may drive thermal adaptive divergences among different geographic populations of *O. oratoria* [25,26], making it a valuable study subject for investigating adaptive evolution. Furthermore, the meat of mantis shrimp is delicious and nutritious, making it a commercially exploited species in coastal waters throughout Japan, Korea, and China [27,28]. However, *O. oratoria* is particularly challenged by short-term environmental stimuli because they are ectotherms and have limited physiologically regulative capacity, making them vulnerable to temperature fluctuations in their surroundings [29,30]. Temporarily cultivating mantis shrimps for post-capture fattening has the potential to achieve higher economic benefits, while the temperature is one of the significant ecological factors that influence their survival during the temporary culture period [31,32]. Although several putative key genes of *O. oratoria* for thermal adaptation were identified through population genomic analysis [33], the dynamic transcriptomic changes in *O. oratoria* exposed to acute heat stress haven’t been unveiled yet. Consequently, it is necessary to gain preliminary knowledge about tissue-specific molecular responses in *O. oratoria* and to explore its dynamically responsive mechanism in the face of transient episodes of high temperatures.

In the present study, we performed a suite of laboratory experiments designed to preliminarily understand the tissue-specific and time-series transcriptional responses of *O. oratoria* to acute heat stress. Initially, *O. oratoria*’s 12 h semi-lethal temperature (12 h LT50) under heat stress was evaluated. Second, experimental *O. oratoria* individuals were treated at an acute high-temperature treatment of 30 °C, which was slightly below the 12 h LT50. In crustaceans, muscle serves as the most abundant amino acid pool; the hepatopancreas is an integrated organ of immunity and metabolism; and the gill has direct contact with the aquatic environment and plays an important role in respiration [34,35,36]. Thus, the time-series samples (0 h, 0.5 h, 6 h, and 12 h) from the abdomen muscle, hepatopancreas, and gill of *O. oratoria* were subjected to high-throughput transcriptome sequencing. With these datasets, we aim to address (i) what unique transcriptional response occurs in different tissues of *O. oratoria*; (ii) how *O. oratoria* responds to acute thermal stress on a transcriptional level over time; and (iii) what is the most prominent response when *O. oratoria* faces an acute thermal challenge. This transcriptional landscape may provide a holistic picture of the molecular responses of crustaceans to short-term thermal exposure and be of significance for *O. oratoria*’s rational development in aquaculture in the future.

## 2. Results

### 2.1. Assessment of 12 h Semi-Lethal Temperature

The survival rates of *O. oratoria* individuals treated with a series of high temperatures in the 12-h semi-lethal temperature (12 h LT50) experiment were summarized (Appendix A). The results showed that *O. oratoria* could not survive at 35 °C, and the survival rates at 31 °C and 33 °C were 87.5% and 50%, respectively. According to the Probit analysis, the 12 h LT50 of *O. oratoria* under heat stress is 32.54 °C. Therefore, we used 30 °C, which is slightly lower than 12 h LT50, as the acute heat stress temperature in the subsequent experiments (Figure 1A).

### 2.2. RNA-Seq Reads

A total of 893,144,074 raw reads were generated from 36 RNA-Seq libraries, and 855,143,123 clean reads were obtained after removing low-quality raw reads. The number of clean bases was approximately 256.54 Gb, and the amount of mean clean data in each RNA-Seq library was 7.13 Gb. The GC content ranged from 42.77% to 49.69% (Appendix A). Over 90% of the bases in all samples exceed Q30, indicating the high quality of the sequencing data. High-quality clean reads were then mapped to the FL transcriptome sequence of *O. oratoria*, and the average mapping rate of 36 libraries was 73.41%.

### 2.3. Tissue-Specific Response to Heat Stress

We investigated patterns of all transcripts’ expression levels with the hierarchal-clustering expression profile analysis and the principal component analysis. The results showed that samples of the same tissue clustered together regardless of heat stress time points (Appendix A), suggesting a general tissue-specific expression pattern. In this study, HS-0h samples of each tissue (muscle: Mus-0h; hepatopancreas: Hp-0h; gill: Gi-0h) were used as the control group and compared to samples of other time points (0.5 h, 6 h, and 12 h) to screen for differentially expressed transcripts (DETs). The number of DETs under different tissue/time conditions is shown in Figure 1B. After removing duplicates from all the identified DETs, a total of 6914 DETs (3395 up-, 3066 down-, and 453 anti-directionally-regulated) were identified. When considering a single tissue, a total of 2126 DETs (1323 up-, 795 down-, and 8 anti-directionally regulated), 2992 DETs (1603 up-, 1314 down-, and 75 anti-directionally regulated), and 3654 DETs (1697 up-, 1934 down-, and 23 anti-directionally regulated) were selected in muscle, hepatopancreas, and gill, respectively (Figure 1C and Appendix A). During the whole heat stress period, gills had the largest number (3654) of DETs. Notably, gills had an overall proportion of 62.3% (1934/3066) in down-regulated transcripts under heat stress, as well as 59.3% (926/1561), 54.6% (848/1553), and 61.4% (1098/1787) at 0.5 h, 6 h, and 12 h, respectively (Appendix A). In addition, according to the Kyoto Encyclopedia of Genes and Genomes (KEGG) enrichment results, the number of pathways enriched by up- and down-regulated transcripts in gills at 0.5 h is the largest (Appendix A). On the other hand, a greater number of functionally upregulated transcripts (Appendix A) and more significantly enriched pathways were induced in the hepatopancreas at 6 h (Appendix A), such as immune activity as well as cell signaling.

A heatmap based on the expression levels of 6914 DETs indicated an obvious tissue-specific expression pattern (Figure 2A). A total of 1262, 1755, and 2298 DETs, which were expressed exclusively in one tissue, were identified as tissue-specific transcripts (TSTs) in the muscle, hepatopancreas, and gill, respectively. As shown in Figure 2B, Gene Ontology (GO) enrichment analysis of TSTs in muscle indicated the most significantly enriched GO terms were related to the processes of protein degradation and chitin metabolism, such as “proteolysis (GO:0006508)” and “chitin metabolic process (GO:0006030)”. The top significant GO terms enriched in the hepatopancreas were related to the functions of lipid transport and catalysis, such as “lipid transport (GO:0006869)” and “fucosyltransferase activity (GO:0008417)”. In the gill, the most significant GO terms were enriched in terms of transport, energy metabolism, and cellular structure, including “transmembrane transporter activity (GO:0022857)”, “gluconeogenesis (GO:0006094)”, and “troponin complex (GO:0005861)”.

### 2.4. Overall Response and HSP Families

To gain a comprehensive understanding of expression profiling during heat stress, we performed trend analysis with 6914 DETs. In total, 10 cluster profiles were used to summarize the expression patterns, of which three were statistically significant (Figure 3A). Notably, 916 transcripts were significantly clustered in profile 9 and constantly upregulated, indicating these transcripts were involved in responses during the whole process to acute heat stress. KEGG enrichment analysis revealed that “protein processing in the endoplasmic reticulum (ko:04141)” (q = 6.1 × 10^−3^) was the only significant pathway (Figure 3B)**,** suggesting that protein processing in the endoplasmic reticulum (ER) was a prominently acute thermal response in *O. oratoria* independent of exposure time. Interestingly, we found that 26 of 54 transcripts significantly enriched in this pathway were identified as members of the heat shock protein (HSPs) family. Therefore, a total of 69 DETs encoding HSPs were screened out, among which 61 (88.4%) transcripts exhibited a constantly increasing trend. To further identify which HSP subfamilies play vital roles in the acute thermal tolerance of *O. oratoria*, these transcripts encoding HSPs were subjected to heatmap analysis based on the expression level and its variation (log_2_FC). Of particular note, we found that most members of the HSP60/HSP70/HSP90 subfamilies were consistently upregulated during the continued heat shock in all tissues, with the HSP60 subfamily showing a prominently high expression pattern (Figure 3C). 

### 2.5. Time-Resolved Response to Acute Heat Stress

When considering DETs in each tissue, the expression profiles showed RNA-seq samples were all divided into three categories (0 h, 0.5 h, and 6 h–12 h) according to heat stress time points (Appendix A), indicating gene expression may have shifted over time. As expected, the number of unique DETs after removing duplicates tended to decrease with heat stress time, from 3540 DETs (1884 up-, 1561 down-, and 95 anti-directionally regulated) at 0.5 h to 3430 DETs (1848 up-, 1553 down-, and 29 anti-directionally regulated) at 6 h and 3297 DETs (1485 up-, 1787 down-, and 25 anti-directionally regulated) at 12 h (Figure 1D and Appendix A). There was a poor overlap among the DETs at each time point, and only 724 (10.5%) transcripts were detected throughout the heat stress process, indicating the dynamic transcriptome changed during the continued heat shock. Taking pairwise comparisons into account, the most (1776, 25.7%) shared heat-responsive transcripts were observed between 6 h- and 12 h-heated groups (Figure 1D).

Weighted gene co-expression network analysis (WGCNA) revealed five modules (lightgreen, black, salmon, darkred, and tan) as heat-responsive transcript clusters that have highly correlative expression profiles with target tissues over heat stress time (Gi-0.5h, Hp-0.5h, Gi-6h, Hp-6h, and Mus-12h) (Figure 4A). The hub transcripts in each target module were also identified, as shown in Figure 4B–F. The functional enrichment analysis of DETs and heat-responsive transcript clusters produced transcriptome-dynamic results based on exhaustive functional annotation information.

At the onset of heat stress at the 0.5 h timepoint, a batch of upregulated transcripts in the gill samples were significantly enriched in the pathways related to cell signaling (e.g., MAPK signaling pathway (ko:04010), Appendix A), including organic cation transporter (ORCT), mitochondrial carrier family MFRN2, Pyrexia, NF-kappa-B p110 (Rel), NF-kappa-B inhibitor (Cact), MAP kinase-interacting serine/threonine-protein kinase 2 (MKNK2), solute carrier (SLC) family members, and transient receptor potential (TPP channels) members (Appendix A). Among them, ORCT, MFRN2, and Pyrexia were further identified as hub transcripts in the Gi-0.5h module (light green, r = 0.85, *p* = 4 × 10^−11^, Figure 4A) (Table 1). Other upregulated transcripts expressed in the gill were significantly enriched in immune-related pathways, such as the IL-17 signaling pathway (ko:04657) and the NOD-like receptor signaling pathway (ko:04621) (Appendix A). Transcripts clustered in the Gi-0.5h module were also significantly enriched in serine-type endopeptidase activity (GO:0004252) associated with immune response (Appendix A). In addition, a widespread upregulation of transcripts encoding proteinase/proteinase inhibitors, and cytokines was observed at 0.5 h (Appendix A). Notably, Astakine (ASTA), PI-stichotoxin-Hcr2e (VKT #1), and Kunitz-type serine protease inhibitor (VKT #2) were identified as the hub transcripts in the Gi-0.5h module, all of which were upregulated at this timepoint (Table 1), suggesting their key immune functions in the early responsive network. For transcripts in the Hp-0.5h module (black, r = 0.76, *p* = 1 × 10^−7^) (Figure 4A), the GO analysis detected an enrichment associated with lipid transport activities (GO:0008061) (Appendix A). Several proteins actively participating in lipid activities encoded by hub transcripts were of high interest, including UDP-glucuronosyltransferase 2B9 (UGT2B9), β-1,3-glucan-binding protein (BGBP), and Apolipophorin (APLP) (Table 1). We also noticed that early-responsive transcripts in the gill and muscle were associated with the cytoskeleton (actin, myosin, troponin, tubulin, and microtubule) (Appendix A). However, transcripts encoding tubulin and microtubule in the gill were upregulated, whereas those encoding actin, myosin, and troponin were downregulated in both tissues. Intriguingly, the functional analysis of upregulated DETs also detected an enrichment with the HIF-1 signaling pathway (ko:04066) (Appendix A), which likely reflected alterations in oxygen availability at high temperatures. Another molecular response recorded at 0.5 h was the marked repression of a high number of transcripts coding for key enzymes associated with aerobic metabolic processes in all studied tissues, including malate dehydrogenase (MDH), NADH dehydrogenase (ND), and ATP synthase (Appendix A).

At the 6 h timepoint, enriched GO terms related to ER homeostasis were revealed in both the Hp-6h module (darkred, r = 0.68, *p* = 6 × 10^−6^, Figure 4A) and the Gi-6h module (salmon, r = 0.74, *p* = 3 × 10^−7^, Figure 4A), corresponding to protein disulfide isomerase activity (GO:0003756) and endoplasmic reticulum (GO:0005783), respectively (Appendix A). Concordantly, we identified transcripts encoding Endoplasmic reticulum oxidoreductin-1 alpha (Ero1) as hub transcripts in the Hp-6h module and transcripts encoding Protein disulfide-isomerases (PDIs), Endoplasmin (HSP90B), and Endoplasmic reticulum lectin 1 (XTP3B) as hub transcripts in the Gi-6h module (Table 1). In addition, several translation-related GO terms, including aminoacyl-tRNA ligase activity (GO:0004812), ligase activity, forming aminoacyl-tRNA and related compounds (GO:0016876), and tRNA aminoacylation for protein translation (GO:0006418) (Appendix A), were significantly enriched in the Gi-6h module, suggesting that 6-h heat stress exposure may have induced the change in protein synthesis in *O. oratoria*. In agreement with this, we found that a bunch of transcripts encoding eukaryotic translation initiation factors (EIFs), elongation factors (EFs), and small nuclear ribonucleoproteins (SNUs) were upregulated at 6 h (Appendix A). Moreover, among the transcripts that were upregulated at 6 h, two pathways related to protein ubiquitination, the proteasome pathway (ko:03050) and the ubiquitin-mediated proteolysis pathway (ko:04120), were significantly represented (Appendix A), suggesting that protein degradation by the ubiquitin-proteasome system (UPS) was activated in *O. oratoria* during acclimation to high temperatures. On the other hand, the transcripts that were downregulated at the middle stage were significantly enriched for the TCA cycle pathway (ko:00020) and Pyruvate metabolism pathway (ko:00620) (Appendix A), indicative of a hypometabolism triggered by heat stress.

At the end of the experimental exposure (12 h), the continued reduction of metabolic activity was evidenced by the downregulation of transcripts that were enriched in metabolic pathways, the TCA cycle pathway, and the Pyruvate metabolism pathway (Appendix A). Similar to the upregulated DETs at 6 h, the two upregulated pathways associated with protein ubiquitination mentioned above were significantly enriched by 590 transcripts that were upregulated at 12 h (Appendix A), and 48 related DETs were mapped to these two pathways (Appendix A). Consistently, we found that 115 transcripts were clustered into the Mus-12h module (tan, r = 0.62, *p* = 5 × 10^−5^, Figure 4A) and were overrepresented with endopeptidase activity (GO:0004298) and proteasome complex (GO:0005839) (Appendix A). Among these transcripts, 34 were found to encode the 26S proteasome complex, a multiprotein complex involved in the ATP-dependent degradation of ubiquitinated proteins. Of note, three transcripts encoding 26S protease regulatory subunit 6B (PRS6B), 26S proteasome regulatory subunit 8 (PRS8), and Proteasome subunit alpha type-1 (PSA1), all of which are components of the 26S proteasome complex, were identified as hub transcripts in the Mus-12h module and showed increased expression levels (Table 1). Therefore, their high association with other transcripts in the Mus-12h module and upregulated expression levels highlight the importance of protein degradation during the late stage under acute heat stress (Figure 4F).

### 2.6. Validation of RNA-Seq Results by qRT-PCR

RNA-Seq results were validated by analyzing the relative expression of eight DETs (PDI6, CLOT, CCT6, HSP90A, HSPA5, PSMC3, ATP6V0A1, and ORCT) involved in the heat stress response of *O. oratoria*. As shown in Appendix A, a significantly good concordance (R^2^ = 0.736, *p* < 0.0001) was observed between the change in expression values of these DETs obtained by RNA-Seq and qRT-PCR experiments, thus supporting the reliability of the data obtained from our transcriptome analysis.

## 3. Discussion

Generally, the cellular processes that respond to environmental perturbations are transcriptionally coordinated in magnitude and time [13,37]. Here, we suggest this is reflected in the transcriptome responses of *O. oratoria* to acute heat stress. Our transcriptomic analyses documented a higher response to heat stress in the first 0.5 h after the temperature upshift, suggesting that a short period of acute stress is enough to elicit regulatory responses. When acute heat stress persisted, gene expression levels tended to settle into a steady state. The function of upregulated pathways at HS-12 h was similar to that at HS-6 h, but the type and number of pathways were reduced at 12 h. These observations seem to indicate that a fast and robust change in gene expression was triggered once *O. oratoria* was exposed to acute heat stress, while the molecular responsive mechanisms may be biphasic; that is, the proactive responsiveness of *O. oratoria* under acute heat stress probably differed between the early and mid-long stages.

Cells must receive signals outside their boundaries and process information quickly to orchestrate subsequent cellular responses when exposed to environmental stress [38,39]. In organisms, the intrinsic properties of cell signaling depend on the interrelationship between lipids and proteins, which is mainly established through membrane transporter proteins [40]. In this study, a large set of transcripts encoding signaling molecules were upregulated once exposed to thermal stress, suggesting that the first step in responding to acute heat stress in *O. oratoria* depends on the ability to detect temperature changes. We also noted that Pyrexia, a transient receptor potential (TRP) channel endowing tolerance to high temperature in *D. melanogaster* [41], was identified as the hub transcript in the Gi-0.5h module, supporting the idea that TRP channels may be activated by acute heat stress and play an important role in the thermo-sensation of *O. oratoria* during the early stage [42,43]. Meanwhile, the upregulated expression of transcripts encoding SLC influx transporters illustrates that the enhanced transmembrane transport process may have contributed to cell signaling in *O. oratoria* after acute thermal exposure by accelerating protein-lipid bilateral exchange. Therefore, one of the early responses to acute thermal challenge in *O. oratoria* was the enhancement of recognition of external stress and transport of substrate, just as in the “alarm reaction” proposed by Selye (1950) [44].

Although invertebrates lack a true adaptive immune response, they could protect themselves from changeable environmental conditions via a series of potent innate immune responses [45]. At 0.5 h, broad innate immune responses of *O. oratoria* were invoked through the upregulation of proteinase/proteinase inhibitors, and cytokines in the gills. This observation may suggest that *O. oratoria* secreted cytokines to recruit immune cells to infection sites and activated the proteolytic cascade, an important component of the invertebrate immune system, as a defense strategy against acute thermal stress [46,47]. There is a special class of immune molecules that can perform the dual roles of lipid transport and innate immune response [48,49,50,51]. In our study, two hub transcripts encoding this class of lipid-associated immune substances, BGBP and APLP, were identified in the Hp-0.5h module. This is due to the related role of the hepatopancreas in lipid storage [50,51,52] and further supports the robust immune system activity of *O. oratoria* in the early period of acute heat stress treatment. Nonetheless, these candidate transcripts encoding immune molecules should be verified experimentally in the future to determine their functional association with acute thermal resistance in *O. oratoria*.

The cytoskeletal organization is known to undergo profound transformation under thermal stress [53], and cells may change different cytoskeletal networks when exposed to stress [54,55]. At the beginning of heat stress (0.5 h), the expression of transcripts encoding the cytoskeleton elements (e.g., actin, myosin, and troponin) in the gills and muscles of *O. oratoria* was significantly decreased. Interestingly, some other transcripts related to the cytoskeleton, such as tubulin and microtubule-associated proteins, were upregulated. This observation seems to indicate that either impaired or remodeled cellular structure may have occurred in *O. oratoria* during early responses to heat stress. Similar findings that different groups of cytoskeleton-related transcripts showed an opposite direction of expression regulation have been demonstrated in other aquatic animals, such as sea squirts exposed to acute heat stress [56] and the banana shrimp *F. merguiensis* under ammonia stress [57]. Further morphological investigation using microscopic techniques such as transmission electron microscopy would be necessary to examine the ultrastructural changes of the gills under heat stress in *O. oratoria*.

Several expected signals of acclimation to acute high temperatures were apparent in *O. oratoria* when acute thermal stress was prolonged. These signatures include the regulation of the processes of protein synthesis and degradation. Maintaining protein turnover plays an essential role in many biological processes, including resistance to environmental pressure. In eukaryotes, the most common mechanism for degrading intracellular proteins is via the ubiquitin-proteasome pathway in the ER [58,59,60]. In this study, the importance of ER homeostasis in the heat stress response was emphasized by the upregulation of many genes (e.g., Erol, PDIs, HSP90B, and XTP3B) that are critical for organisms to accommodate the unique features of the ER environment [61,62,63]. Increased expression of a set of transcripts enriched in protein synthesis function discloses an increase in the translation speed of *O. oratoria* during the mid-stage of heat stress. On the other hand, the evidence that protein ubiquitination and the 26S proteasome complex were positively regulated suggests *O. oratoria* may maintain a balance in protein metabolism by regulating protein abundance and amino acid recycling as a result of acclimation to the continued acute heat stress. 

The mismatch between metabolic oxygen demand and oxygen availability in water during acute heat stress is a known process in aquatic organisms [64,65]. Our observation of activation of the HIF-1 pathway in gills at the early stage and all down-regulated expression of key genes involved in aerobic metabolic processes during the entire experiment may suggest a trade-off limiting oxygen consumption. Another heat-stress effect observed in *O. oratoria* at the middle-late stages was a widespread downregulation of expression in a variety of metabolic pathways. This could be explained by the fact that numerous aquatic organisms respond to the dilemma posed by metabolic oxygen demand through an across-the-board reduction of metabolic activity at nearly lethal temperatures [66]. The observed downregulation of metabolic processing may suggest metabolic compensation as a possible adjustment mechanism for *O. oratoria* to cope with heat stress, as has been implicated in yeast [67] and other invertebrates [68,69,70]. Further studies based on proteomic data are required to clarify the protein-level mechanisms of this plastic response.

During the whole heat-resistant process, the consistently upregulated HSPs detected after heat shock showed evidence of the high transcriptome plasticity of *O. oratoria* to cope with acute heat stress. HSPs are known to play protective roles as molecular chaperones and are involved in cellular survival as well as stress tolerance in response to thermal challenges [71,72] Among different HSP families, high molecular weight HSPs (HMW HSPs) are of great significance in helping synthesized proteins fold and modulate transcription factors as well as protein kinases [73]. In the present study, HMW HSP family members (HSP60, HSP70, and HSP90) were upregulated in all tissues during the entire thermal exposure process, highlighting their importance for maintaining heat resistance in *O. oratoria*. Moreover, transcripts encoding HSP60 showed predominantly higher expression levels, similar to the case observed in *L. vannamei* under formalin stress [74]. It has been reported that HSP60 participates in intrinsic immune and stress responses under different environmental stressors and might perform important functions indispensable to a variety of physiological processes [75]. It is therefore conceivable that HSP60 may exercise multiple functions, such as immune resistance in *O. oratoria,* when exposed to acute heat shock. 

In addition to the time-resolved changes in the transcriptome, a tissue-specific pattern was identified in the expression profiles of all transcripts as well as DETs, showing that tissue type was an important factor affecting transcriptome results. The GO enrichment analysis of TSTs revealed that the unique adaptation of muscle, hepatopancreas, and gill in *O. oratoria* was related to protein degradation, lipid transport, and energy metabolism, respectively. This is coincident with our observation in the time-resolved response process, that is, the modification of lipid activity dominated the molecular response of the hepatopancreas to heat stress at 0.5 h, the metabolic activity of the gill at 6–12 h, and protein degradation mediated by the UPS system of muscle at 6–12 h. These specific acute thermal responses of each tissue in *O. oratoria* reflect their physiological functions in crustaceans [35,52,76]. Furthermore, there were the most pathways enriched by up- and down-regulated transcripts in the gills at the early stage. Unlike decapod shrimps, the gill of *O. oratoria* is in the abdomen and completely exposed to the water environment. The large and rapid molecular responses in the gill indicate its important role in the early resistance of mantis shrimp to acute thermal stress. On the other hand, the hepatopancreas of *O. oratoria* seems to perform a critical heat-responsive function during the mid-stage. It is evidenced by the fact that upregulated transcripts and significantly enriched pathways were largely induced in the hepatopancreas at 6 h, which in turn highlights its importance in the heat-resistant process in *O. oratoria*. Collectively, the tissue-specific response of *O. oratoria* to acute heat stress in combination with previous observations in other crustaceans (e.g., the Pacific white shrimp *L. vannamei* [77]) concur in suggesting a cooperative heat-responsive network in different tissues of crustaceans.

## 4. Materials and Methods

### 4.1. Sample Collection and High-Temperature Stress Design

Three hundred experimental adult male *O. oratoria* individuals (weighing around 35.5g) were collected from offshore Qingdao, China. All *O. oratoria* individuals were acclimatized in recirculating seawater (temperature: 20 ± 0.5 °C; salinity: 31 psu; ph: 7.5; DO: 8 ± 1 mg/L) for 7 days and fed freshly shelled clams twice a day until the day before the experiment. The photoperiod followed the natural day-light cycle, and one-third of the water was changed daily.

The static method was applied to measure a 12 h Semi-lethal temperature (12 h LT50) of *O. oratoria*. We set different constant temperature conditions (20, 25, 27, 29, 31, 33, and 35 °C) in seven water tanks. Three biological replicates were set for each temperature condition, and eight experimental individuals were placed in each replicate. In total, there were 168 individuals used in the 12 h LT50 assessment of *O. oratoria*. For all experimental groups, three replicates were set up, with eight individuals in each replicate. Criteria for death were loss of balance, submersion in the water, and unresponsiveness to external stimuli, and the survival rates of experimental individuals in each treatment group were observed for 12 h to avoid the effect of circadian rhythms on the stress response of *O. oratoria*. The 12 h LT50 of *O. oratoria* was calculated using a Probit model in SPSS 21.0 software (SPSS Inc., Chicago, IL, USA). 

Subsequently, the heat stress treatment was conducted at a high temperature of 30 °C, which was slightly below the 12 h LT50 (see Results above). Active and healthy Japanese mantis shrimp were randomly allocated into three pre-heated tanks after acclimation. A total of 36 *O. oratoria* individuals were randomly selected during the whole acute heat stress treatment (4 time points (0 h, 0.5 h, 6 h, 12 h) × 3 biological replicates/timepoint × 3 individuals/replicate), and their gills, hepatopancreas, and abdominal muscle were immediately dissected, frozen in liquid nitrogen, and stored at −80 °C.

### 4.2. Total RNA Extraction and Illumina Sequencing

Total RNA was extracted using the TRIzol kit (Thermo Fisher Scientific, Waltham, MA, USA) under the manufacturer’s instructions. The integrity and purity of RNA were measured using an Agilent 4200 bioanalyzer (Agilent Technologies, Santa Clara, CA, USA) and NanoDrop 2000 (Thermo Fisher Scientific, Waltham, MA, USA). Only RNA samples meeting the criteria (RIN^e^ ≥ 6.5; RNA concentration ≥ 40 ng/uL; total amount of RNA ≥ 2 ug) were used for cDNA library construction.

Thirty-six Illumina libraries were constructed using the protocol of the NEBNext Ultra RNA Library Prep Kit (Illumina, San Diego, CA, USA). Briefly, poly (A) mRNA was purified from total RNA using Oligo (dT) magnetic beads and then broken into short fragments using fragmentation buffer to synthesize first-strand cDNA using a random hexamer primer. Second-strand cDNA was then synthesized using DNA polymerase I and RNaseH. The cDNA was subjected to end-repair, phosphorylation, 3′ adenylation, and ligation to sequencing adaptors. Afterward, cDNA libraries were generated by PCR amplification, and the library preparations were sequenced on an Illumina NovaSeq 6000 platform with a 150 bp paired end. To obtain high-quality clean reads, low-quality sequencing reads with sequencing adaptors, low-quality reads (more than 20% bases’ quality score ≤ 5), and reads containing ploy-N (number of ‘N’ > 3) were removed from all raw reads.

### 4.3. Quantification and Functional Analysis of Differentially Expressed and Tissue-Specific Transcripts

The SMRT full-length transcriptome of *O. oratoria* was used as a reference sequence and consisted of 42,735 unigenes with an N50 of 3472 bp, including 33,741 (80.0%) matching nucleotide collection (Nr/Nt) entries (E-value < 1× 10^−5^) [26]. All Illumina clean reads were aligned to the above reference sequence using bowtie2 v2.3.5.1 [78] with parameters: —no-usual —end-to-end -k 200 —no-mixed —no-discordant —gbar 1000. Transcript expression levels of three tissues at each heat stress timepoint were estimated using RSEM [79] with default parameters. Transcripts per Kilobase of exon model per Million mapped reads (TPM) values were used to normalize the reads from RNA-Seq and then log_2_ transformed for correlation analysis. The correlation between each pair of RNA samples was identified using Spearman’s hierarchical clustering in the gplots R package and principal component analysis in the PCAtools R package (R Development Core Team, 2022).

Multiple comparisons were performed to estimate the overall and dynamic transcriptional changes in *O. oratoria* during the acute heat-shock period. Differential expression analysis was implemented using the R Bioconductor package DESeq2 [80]. Transcripts were analyzed from read counts and assigned as DETs with the criteria of corrected *p*-value < 0.05 and |log_2_ fold change (FC)| > 1. A Venn diagram was constructed in the Venn diagram R package [81] to visualize the number of DETs at different time points and in different tissues. To reveal the tissue-specific response of *O. oratoria* to heat stress, we defined DETs exclusively identified in one tissue as TSTs according to the visualization of Venn analyses of DET datasets.

Functional enrichment analyses, including GO and KEGG pathways, were performed to investigate which GO items and metabolic pathways the candidate transcript dataset participated in, respectively. Specifically, GO enrichment analysis of candidate transcripts was implemented in the GOseq R package [82]. For KEGG enrichment analysis, candidate transcript lists were incorporated into the KEGG database to identify genes involved in particular pathways that occur at a higher frequency than would be expected for a randomly selected set of genes, as implemented in the ClusterProfiler R Package [83]. The GO terms and pathways with an adjusted *p*-value *<* 0.05 were defined as being statistically overrepresented.

### 4.4. Trend Analysis

To obtain insights into the transcriptome dynamics of *O. oratoria* during the continued acute heat shock, trend analysis was performed in the STEM v1.3.13 [84] to exhibit temporal transcript expression profiling. Average expression values at each time point were used as the input to generate the soft clusters. Transcripts clustered in the upregulated profile were subjected to KEGG enrichment analysis to further elucidate the detailed molecular responses using the methods described in 4.3. Heatmap analysis for the expression profiling of transcripts encoding specific gene family members was performed and visualized using the Pheatmap R Package (R Development Core Team, 2022).

### 4.5. Weighted Gene Co-Expression Network Analysis (WGCNA)

WGCNA was performed to classify groups of transcripts that interact at the network level during the heat stress process [85]. We screened transcripts with the top 25% Median Absolute Deviation (MAD) and filtered out low-expressed transcripts (TPM < 1) in 36 transcriptome sequencing libraries as effective transcripts for WGCNA analysis. The soft threshold power β = 14 was selected to construct a scale-free network. Effective transcripts in the 12 tissue-time groups were hierarchically clustered, and the resulting gene tree map was cut by dynamic cutting to obtain gene modules. The eigengene value for each module was calculated using all transcripts within each module and correlated with each sample. Only Cor values greater than 0.6 and *p* < 0.05 were identified as target modules. Transcripts within the highly related modules with module membership (MM) values > 0.7 were subjected to GO enrichment analysis. The top 3-6 heat-stress-induced transcripts with both high MM values and gene significance (GS) were identified as hub transcripts in each target module. Cytoscape 3.9.1 software [86] was used to construct the visualization network for target modules.

### 4.6. qRT-PCR for DETs Validation

To validate the reliability and accuracy of RNA-Seq analysis, SYBR Green-based quantitative real-time PCR (qRT-PCR) was performed to detect the relative quantification of eight heat-responsive selected DETs with high expression levels (TPM ≥ 50). The expression level of each transcript was validated using the tissue samples where the transcript was detected as DETs originally in transcriptome sequencing, and there were three biological replicates for each DET and two technical repetitions per biological replicate. All primers designed by Premier 5.0 were listed in Appendix A. Moreover, the 18S rRNA was used as an internal standard [25]. qPCR was conducted using a 20ul reaction system with the following program: denaturation at 94 °C for 2 min; 45 cycles of 94 °C for 15 s; annealing temperature of 60 °C for 30 s. The melting curve was used to determine the specificity of the PCR product. Additionally, relative expression levels were analyzed by the 2^−ΔΔCt^ method [87]. The results were subjected to a one-way analysis of variance (ANOVA) using SPSS 21.0 software, and the cut-off *p*-value of 0.05 was considered statistically significant.

## 5. Conclusions

Our study revealed a tissue-specific and time-resolved gene expression pattern in *O. oratoria* under acute heat stress. Tissue type was an important factor affecting transcriptome results, and the unique adaptation of muscle, hepatopancreas, and gill of *O. oratoria* was related to protein degradation, lipid transport, and energy metabolism, respectively. A biphasic protective responsiveness of *O. oratoria* under acute heat stress developed from the early response of signal transduction, immunity, and cytoskeleton reorganization to the response dominated by protein turnover and energy metabolism at the mid-late stages (Figure 5). HMW and HSP family members stand out as important molecular components for maintaining heat resistance in *O. oratoria*. This study greatly improves the current knowledge of gene expression activity in marine crustaceans under acute heat stress across different tissues at fine temporal resolution. Several lines of investigation that use comparative approaches (e.g., transcriptomic and proteomic analyses) are necessary to elucidate adaptive strategies to high temperatures, and further studies would be needed to consider the multi-stressor effects of environmental alterations occurring with climate change on marine crustaceans.

## Figures and Tables

**Figure 1 ijms-24-11936-f001:**
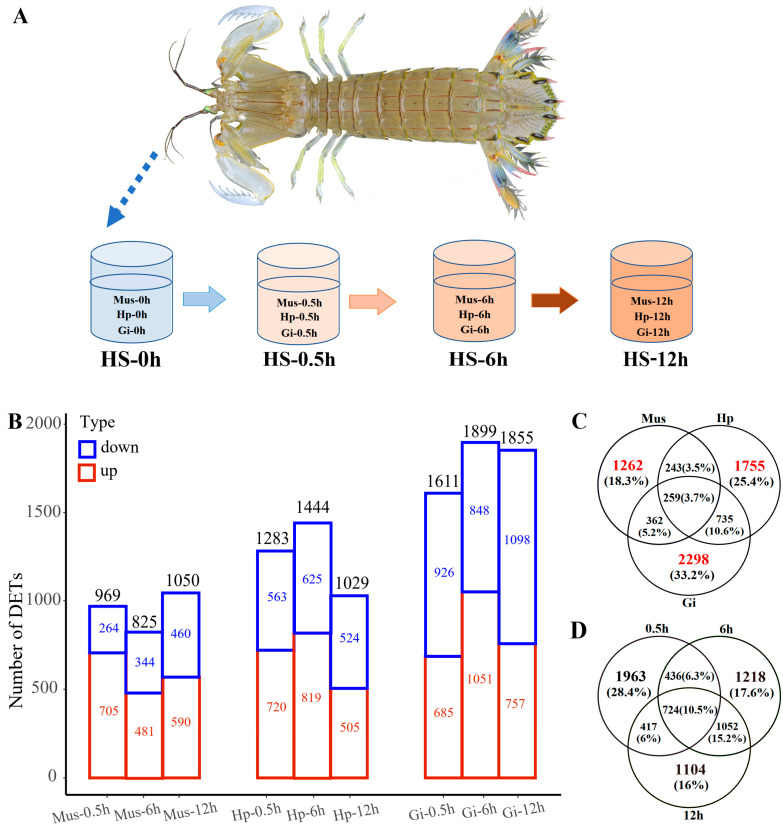
Experimental design (time and tissues) during heat stress laboratory trials (**A**) and the DETs identified in the heat stress transcriptome of *O. oratoria*. (**B**) Number of DETs among three tissues at different heat stress time points. The Venn diagram of DETs among (**C**) three tissues and (**D**) three heat stress time points.

**Figure 2 ijms-24-11936-f002:**
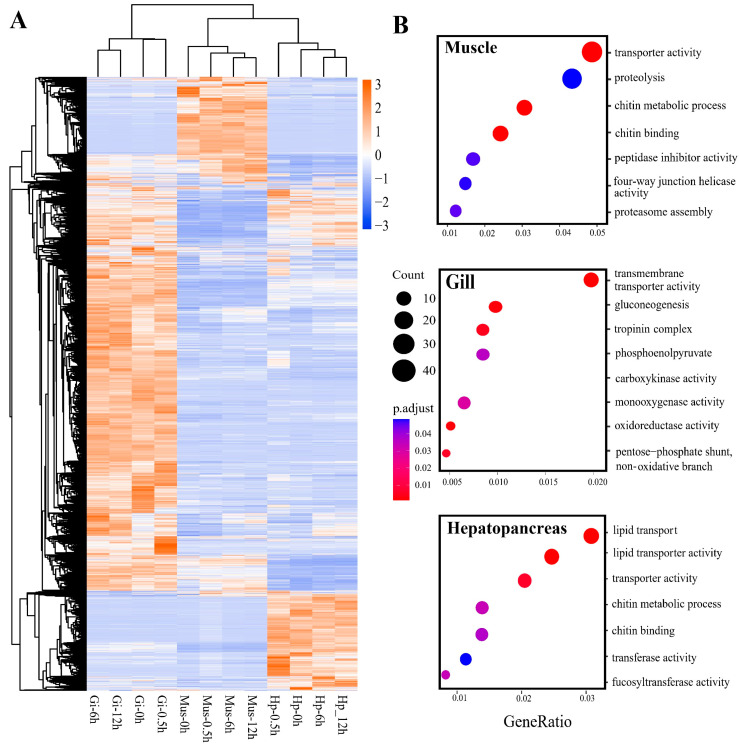
Tissue-specific expression pattern of *O. oratoria* revealed by (**A**) hierarchical clustering analysis based on all DETs and (**B**) GO enrichment analysis of exclusively differentially expressed transcripts (TSTs) in muscle, gill, and hepatopancreas, respectively.

**Figure 3 ijms-24-11936-f003:**
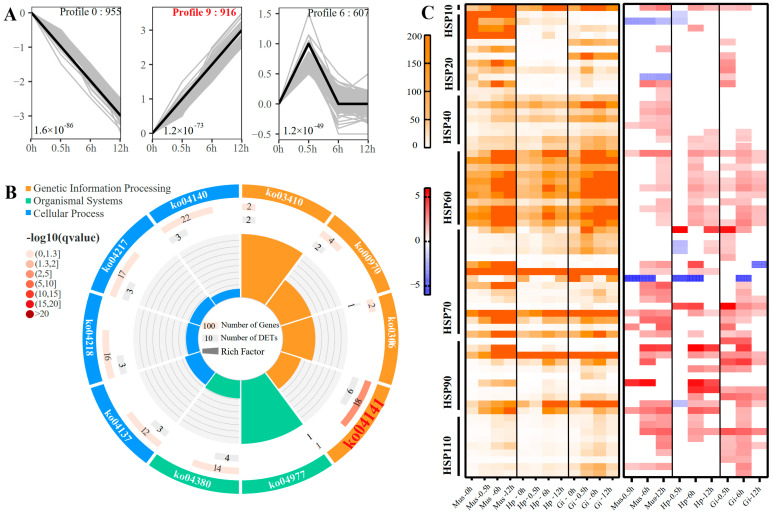
(**A**) Significant transcript expression profiles in trend analysis of *O. oratoria*. The number of transcripts and the enriched significance of each profile are shown at the bottom and top of each frame, respectively. The black lines represent the model expression pattern. (**B**) KEGG enrichment analysis of the transcript set in profile 9. (**C**) Heatmap of transcripts encoding heat shock proteins based on the expression level and the values of log_2_FC, respectively. The orange color represents the expression level, and the red and blue colors represent the values of log_2_FC.

**Figure 4 ijms-24-11936-f004:**
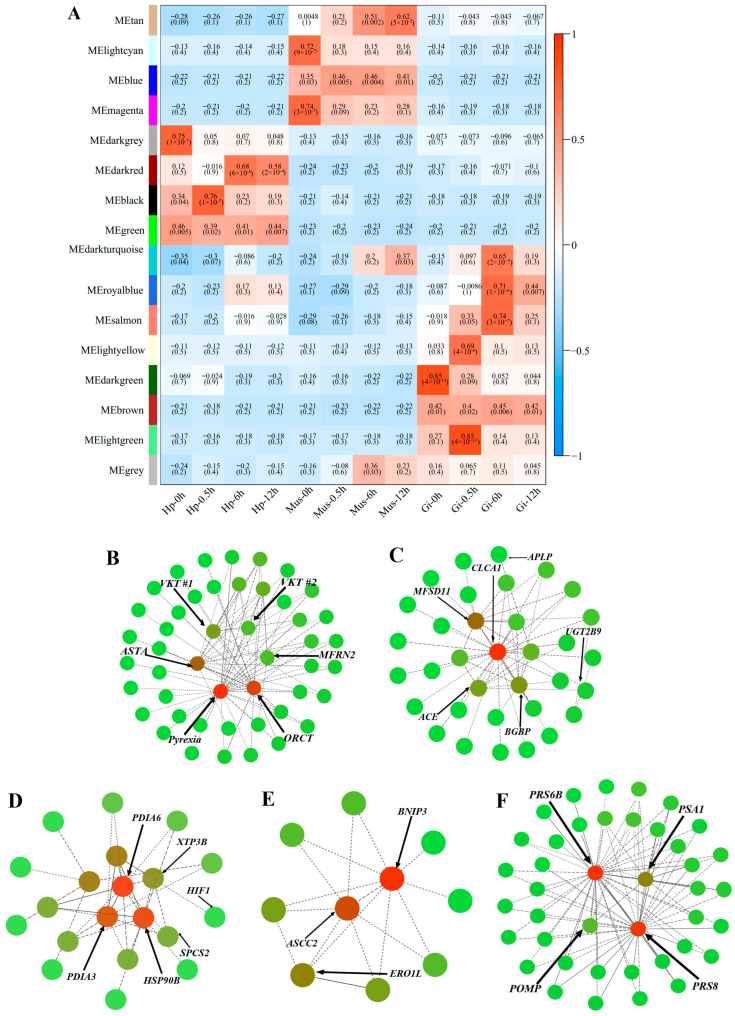
Weighted gene correlation network analysis (WGCNA) of all samples in *O. oratoria*. (**A**) Correlation matrix between modules and samples. Co-expression network was divided into 15 distinct co-expression modules (tan, lightcyan, blue, magenta, darkgrey, darkred, black, green, darkturquoise, royal blue, salmon, lightyellow, darkgreen, brown, and lightgreen). The x-axis represents different samples, and the y-axis represents the name of each module. The *p*-value is presented in each cell. (**B**–**F**) Correlation network visualization of the interactions between the top 3-6 transcripts with the top connectivity as indicated by their high KME (eigengene connectivity) values in five target modules. The nodes represent transcripts, the edges represent intermolecular interactions between adjacent nodes, and the arrows indicate the hub transcripts.

**Figure 5 ijms-24-11936-f005:**
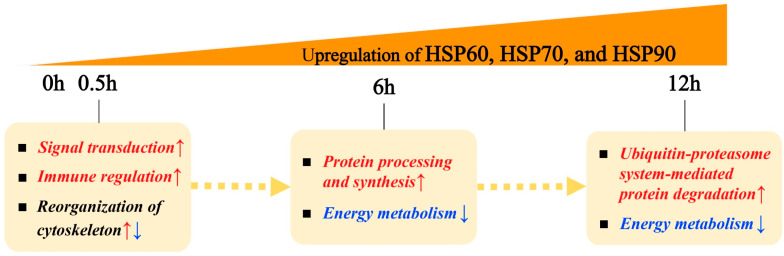
The dominant time-series molecular responses of *O. oratoria* under acute thermal challenge. The up- and down-regulated biological processes after acute thermal exposure are represented by red and blue arrows, respectively.

**Table 1 ijms-24-11936-t001:** Hub transcripts selected from 5 target modules.

Module	Id	UniProt Annotation	log_2_FC			
			Gi-0.5h	Gi-6h	Hp-6h	Mus-12h
lightgreen module & Gi-0.5h	transcript/45325/f10p0/1877	Astakine	1.12			
	transcript/64068/f3p0/610	PI-stichotoxin-Hcr2e	1.07			
	transcript/57382/f8p0/949	Kunitz-type serine protease inhibitor bungaruskunin	1.11			
	transcript/20729/f140p0/3462	Organic cation transporter protein	1.53			
	transcript/5173/f4p0/5336	Mitoferrin-2	1.20			
black module & Hp-0.5h	transcript/4046/f2p0/5700	Angiotensin-converting enzyme				
	transcript/22992/f9p0/3315	Calcium-activated chloride channel regulator 1				
	transcript/48715/f5p0/1617	UDP-glucuronosyltransferase 2B9				
	transcript/1109/f2p0/6853	β-1,3-glucan-binding protein				
	transcript/43086/f5p0/2047	Apolipophorin				
salmon module & Gi-6h	transcript/59269/f2p0/853	signal peptidase complex subunit 2		1.41		
	transcript/42720/f2p0/2103	Protein disulfide-isomerase A3		2.49		
	transcript/25284/f114p0/3127	Endoplasmic reticulum lectin 1		1.11		
	transcript/42991/f26p0/2035	Protein disulfide-isomerase A6		3.33		
	transcript/30855/f1028p0/2766	Endoplasmin		2.84		
	transcript/4180/f5p0/5632	Hypoxia-inducible factor 1-alpha		3.93		
darkred module & Hp-6h	transcript/43279/f19p0/2039	BCL2/adenovirus E1B 19 kDa protein-interacting protein 3			2.02	
	transcript/43422/f3p0/2057	Activating signal cointegrator 1 complex subunit 2			2.04	
	transcript/39969/f23p0/2280	Endoplasmic reticulum oxidoreductin-1 alpha			2.94	
tan module & Mus-12h	transcript/49905/f14p0/1510	26S protease regulatory subunit 6B				1.42
	transcript/51481/f14p0/1401	26S protease regulatory subunit 8				1.31
	transcript/56850/f5p0/989	Proteasome subunit alpha type-1				1.03

## Data Availability

All raw data had been deposited on the NCBI Sequence Read Archive (SRA) website with the accession number PRJNA944923.

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
