# Peer review of "Time-Course and Tissue-Specific Molecular Responses to Acute Thermal Stress in Japanese Mantis Shrimp Oratosquilla oratoria"

_ijms, 2023, doi:10.3390/ijms241511936_

Round 1
Reviewer 1 Report
It would be good to give a photograph of the studied shrimp Oratosquilla oratoria in Materials and Methods and describe in more detail the practical value of this species
Reviewer 2 Report
The present manuscript ID: ijms-2460856 by Zhang et al. investigated tissue-specific and time-resolved gene expression patterns of O. oratoria to acute heat stress. The presentation of this manuscript should be improved especially on the quality of figures and methodology description. I have several comments which can be helpful to improve the present version. The manuscript was written well, and the topics were also interesting.
The title is too cumbersome. What does landscape indicate?
Line 84- the objectives must be clearly defined
Section 4.1: It is unclear on “directly exposed to a series of higher water temperatures (25, 27, 29, 31, 33, and 35 °C) from the acclimation water temperature (20 °C)”
Section 4.6- The calculation used for DEG is Transcripts Per Kilobase of exon model per Million mapped reads (TPM) and for qPCR is 2-ΔΔCt. Although the author considered validating TPM using qPCR which measures target gene fold change at the given time point. How is the calculation similar between RNA-seq and qPCR? It would be better to select DEG by fold change >2 or 4.
The author should mention how was the DEG selected for validation. It is advisable to select DEG from the most enriched pathway using GO or KEGG for validation.
None of the figures from Fig 2, fig 4 and fig5 is readable. The presentation is not acceptable. Authors must present high-resolution figures.
There are too many figures and tables presented in this manuscript and the author should consider some of them to be moved to the supplementary section.
The manuscript is well-written, and minor english revisions are required.
Reviewer 3 Report
In my opinion, the manuscript submitted by Zhang et al. is a well-designed and written, and executed study that deserves publication. The authors presented interesting compiling study showing the effect of acute thermal stress in mantis shrimp at a molecular level. They thoroughly describe the process response of heat stress over the time exposure tested where LT50 is reached and provide a great discussion of the progression of the biological processes effected. I really liked how Figure 6 shows and capture the main thesis of this study and would be extremely helpful for stakeholders in the aquaculture industry and non-molecular ecologist.
I could not find much room for improvement across the manuscript, but here some improvements I can suggest:
In the result section species scientific name needs to be in italics, both in main text and figures.
Section 2.2.12 subtitle is not very informative, is basically acronyms or abbreviations, suggest changing for a more informative subtitle for this section.
In the section 2.3 Tissue-Specific response it was hard to extract the results described in the main text from the Figure 2. Seems like the number do not match or these results need to be better explained to avoid confusion. For example, in line 127-135 3,654 DETs ( 1700 up- and 1931 down-regulated but in Figure 2A the sum of number of DETs in gill up-regulated is 2,490 and down-regulated is 2.872, nor the Venn diagram show 62.3% transcripts in gill tissue (line 133-135). Overall, Figure 2 is good but found it confusing and hard to match with the main text.
Figure 3 -5 include species name.
Figure 4C, log2FC values should be red and blue, in figure only mentioned red.
In methods and maybe the first time mentioned in the results, I would describe GO and KEGG for consideration to non-expert readers.
